# Pancreatic Enzyme Replacement Therapy in Pancreatic Cancer

**DOI:** 10.3390/cancers12020275

**Published:** 2020-01-22

**Authors:** Raffaele Pezzilli, Riccardo Caccialanza, Gabriele Capurso, Oronzo Brunetti, Michele Milella, Massimo Falconi

**Affiliations:** 1Gastroenterology Unit, San Carlo Hospital, Via P. Petrone, 85100 Potenza, Italy; 2Clinical Nutrition and Dietetics Unit, Fondazione IRCCS Policlinico San Matteo, Viale Camillo Golgi 19, 27100 Pavia, Italy; R.Caccialanza@smatteo.pv.it; 3Clinical Research, Pancreato-Biliary Endoscopy and EUS Division, Pancreas Translational and Clinical Research Center, IRCCS San Raffaele Scientific Institute, Via Olgettina 60, 20132 Milano, Italy; capurso.gabriele@hsr.it; 4Medical Oncology Unit, National Cancer Institute “Giovanni Paolo II”, Viale O. Flacco 65, 70124 Bari, Italy; dr.oronzo.brunetti1983@gmail.com; 5Residency Program in Medical Oncology, University of Verona, Via S. Francesco 22, 37129 Verona, Italy; michele.milella@univr.it; 6AOUI Verona, Sede Policlinico Universitario G.B. Rossi Borgo Roma, P.le L.A. Scuro 10, 37134 Verona, Italy; 7Pancreatic Surgery, Pancreas Translational & Clinical Research Center, IRCCS San Raffaele Scientific Institute, Via Olgettina 60, 20132 Milano, Italy; falconi.massimo@hsr.it

**Keywords:** pancreatic enzyme replacement therapy, pancreatic exocrine insufficiency, pancreatic cancer, chemotherapy, pancreatic resection, nutritional support

## Abstract

Pancreatic cancer is an aggressive malignancy and the seventh leading cause of global cancer deaths in industrialised countries. More than 80% of patients suffer from significant weight loss at diagnosis and over time tend to develop severe cachexia. A major cause of weight loss is malnutrition. Patients may experience pancreatic exocrine insufficiency (PEI) before diagnosis, during nonsurgical treatment, and/or following surgery. PEI is difficult to diagnose because testing is cumbersome. Consequently, PEI is often detected clinically, especially in non-specialised centres, and treated empirically. In this position paper, we review the current literature on nutritional support and pancreatic enzyme replacement therapy (PERT) in patients with operable and non-operable pancreatic cancer. To increase awareness on the importance of PERT in pancreatic patients, we provide recommendations based on literature evidence, and when data were lacking, based on our own clinical experience.

## 1. Introduction

Pancreatic cancer (PC) is an aggressive malignancy and the seventh leading cause of global cancer deaths in industrialised countries [1], the third most common cause of cancer death in the United States [2], and the fourth most common cancer death in Italy [3]. It is expected to become the second most common cancer death by 2030 [4]. The five-year survival rate remains at a mere 9%, as patients with pancreatic cancer seldom exhibit symptoms until reaching an advanced stage of the disease [1]. Despite advancements in disease detection and management, more than 80% of patients receive a diagnosis in the advanced stage of the disease [5]. A similar number of patients suffer from significant weight loss at diagnosis, which can result in severe cachexia [6]. A major cause of weight loss is malnutrition. As shown in Figure 1, both intrinsic (tumour cell-triggered) and extrinsic (host-triggered) factors contribute to malnutrition in PC patients. Tumour cell-triggered factors include: (i) The Warburg effect, characterised by an increased lactate production as a consequence of glycolysis rather than oxidative phosphorylation of glucose, which results in acidosis. This, in turn, leads to an acid-mediated tumour invasion and the impairment of mitochondrial functions in cancer cells, (ii) the production of tumour-specific factors such as islet amyloid polypeptide (IAPP), which contributes to cachexia and the loss of appetite, and (iii) the location of the tumour itself, most often in the head of the pancreas, resulting in the reduction of the secretion of pancreatic enzymes during meals [6]. The latter is referred to as pancreatic exocrine insufficiency (PEI), leading to maldigestion and secondary malnutrition. PEI can also be a consequence of surgery, for instance due to the reduction of glandular tissue following pancreatic resection [7]. Hence, patients may experience PEI before diagnosis, during nonsurgical treatment, and/or following surgery [8].

The impact of PEI and malnutrition cannot be underestimated. Malnutrition can lead to skeletal muscle wasting and fat degradation, longer hospital stays, and increased risk for complications. It reduces the response to the treatment and the patient’s well-being, while increasing the risk of morbidity and mortality in operated and non-operated patients [9,10]. Hence, it is clear that nutritional support is important in patients suffering from PC. The standard treatment for PEI is oral pancreatic enzyme replacement therapy (PERT). Despite the importance of treating PEI and malnutrition in PC patients, studies suggest that treatment is sub-optimal in more than half of patients [11]. One of the reasons for this might be that the nutritional support differs depending on the type of patient: Patients with unresectable pancreatic cancer who undergo the best supportive care may need different nutritional support than patients with locally advanced/borderline resectable (LA/BR) pancreatic cancer during neoadjuvant therapy, while resected patients may have their own specific needs [12]. 

The goal of this position paper is two-fold: First, we aim to create and/or increase awareness on the importance of treating malnutrition and PEI in PC patients. Secondly, we want to provide practical information on the different and appropriate PERT options for the different patient groups.

## 2. Nutritional Support Principles in Pancreatic Cancer

Malnutrition and sarcopenia are well-known factors associated with a limited tolerance to chemotherapy (CT), post-surgical complications, short survival, and poor quality of life (QoL) in PC patients [12,13]. Andreyev et al. showed that patients with weight loss received lower doses of CT and fewer doses, due to the development of more frequent and more severe dose-limiting toxicities [14]. Additionally, continuous weight loss during the third-month induction treatment period, more than weight loss at diagnosis, significantly precluded resection and was an independent factor of shorter overall survival (OS) in unresected patients [15]. Hence, prompt and appropriate nutritional support is fundamental in all PC patients since the early phases of the disease.

Different tools for nutritional screening have been validated in the oncologic setting. These tools allow for the identification of patients at nutritional risk, who are likely to benefit from nutritional therapy [16]. Unfortunately, they are only rarely used [17]. Nutritional counselling is often overlooked in the management of PC patients, but should be part of the first-line treatment in malnourished cancer patients or those at nutritional risk, due to its proven efficacy in increasing protein-calorie intake, body weight, and improving body composition [18]. 

As a general rule, the diet should be as normal as possible, while fat restriction and high-fibre diets should be avoided. Small, frequent, and high-energy meals are generally recommended, as they are easier to digest than large meals in patients with reduced pancreatic secretion [19]. Oral protein–calorie nutritional supplements are required in patients who cannot meet their nutritional requirements despite dietary intervention [18,20]. Fish oil with a high omega-3 content appeared to increase or stabilise weight and improve the appetite of PC patients with cachexia [21]. Additionally, marine phospholipids appear to have a stabilising effect on weight and are slightly better tolerated than fish oil capsules [21]. High doses of L-carnitine showed promising results in patients with cachexia, with improved malnutrition and body composition [22]. However, further investigation is needed to establish the ideal dose and potential long-term effects of these supplements [21]. A recent randomised study showed that, in malnourished advanced cancer patients undergoing CT and receiving nutritional counselling, a three-month supplementation with whey protein isolate resulted in improved body composition, muscle strength, body weight, and reduced CT toxicity [23]. The complexity of effective dietary interventions in this cancer type is amplified by PEI [19]. Therefore, together with dietary counselling, PERT should be included in the nutritional therapy [19,21].

In recent years, calorie-restricted diets and supplements such as vitamins, curcumin, green tea extracts, and aloe have attracted attention for their potential anticancer effects. While definitive conclusions cannot be drawn due to the lack of clinical evidence, many patients adopt complementary anti-tumour strategies aiming to improve efficacy or reduce the toxicity of CT. However, the benefits are uncertain and the risk of additional toxicities or antagonistic interactions with standard therapies is real. Hence, dietary advice should be tailored to the individual patient and “hypocaloric alternative anticancer diets” and “natural nutrients” not supported by clinical evidence avoided [18]. 

In patients who are unable to tolerate sufficient oral food intake (due to anorexia, gastrointestinal symptoms, or physical obstructions, among other reasons), artificial nutrition is recommended [24]. In the case of the intestinal functions of the PC patient being preserved, enteral nutrition is preferred over parenteral as it as efficient as parenteral feeding, but it maintains the gut barrier, has less infection complications, and has lower costs [20]. In cases of severe intestinal insufficiency, parenteral nutrition should be considered to maintain the nutritional status. Indeed, a literature study and several clinical trials reported more long-term benefits of PN in the majority of patients with pancreatic or advanced cancers [21,25,26].

In conclusion, the nutritional status and the survival of PC patients improve upon early nutritional counselling [22]. Therefore, these patients should receive treatment in a multidisciplinary setting, including a nutrition specialist. If this is not possible, oncologists, surgeons, or gastroenterologists should refer the patient to a clinical nutrition specialist to implement prophylactic nutritional monitoring and/or counselling.

## 3. Real Life Diagnostic Approaches of Maldigestion

The most common clinical symptom of PEI is steatorrhea, which is defined as >7 g/day of faecal fat in the stool over 24 h during a 72 h stool test, after being on a diet containing 100 g of fat per day (also called Coefficient of Fat Absorption or CFA). Associated symptoms are abdominal pain, flatulence, and weight loss [27]. As a consequence of malabsorption, patients can develop vitamin deficiencies resulting in secondary symptoms, such as a vitamin K deficiency leading to coagulation abnormalities [28]. It should be noted that steatorrhea only appears late in the disease, when the pancreatic lipase and trypsin levels fall below 5–10% of normal production [29]. The CFA test is considered to be the gold standard for assessing PEI, but is not generally used in clinical practice.

Several other tests are available for diagnosing PEI, both direct tests that measure the direct pancreatic function, and indirect tests that assess the consequence of exocrine insufficiency and evaluate quantitative changes in pancreatic secretion. Direct tests usually require an infusion of hormonal secretagogues to stimulate the release of pancreatic juice in the duodenal lumen. Enzymes and bicarbonate are measured in the collected duodenal fluid [30]. These tests are not standardised—they can use different secretagogues (either secretin or cholecystokinin), different tools to collect the pancreatic juice (double-lumen gastroduodenal tubes or endoscopic aspiration), and different time points for fluid collection [31,32,33]. The tests have good sensitivity, but are invasive, time-consuming, expensive, and not useful in monitoring the response to PERT. 

The most commonly used indirect test is the faecal elastase 1 (FE-1) test [34,35]. Elastase-1 is a proteolytic enzyme produced by pancreatic acinar cells, which passes through the gut with only a minor degree of degradation, being, therefore, quantifiable in faecal samples. Elastase-1 is highly stable in faeces for up to one-week at room temperature, and for one-month when stored at 4 °C, making conservation simple. A concentration of FE-1 <200 µg/g in the faeces is considered abnormal. The available kits measure the amount of FE-1 via either a monoclonal or polyclonal enzyme-linked immunosorbent assay (ELISA). The sensitivity of the monoclonal ELISA for mild, moderate, and severe PEI in patients with PC has been estimated to be 63%, 100%, and 100%, respectively [36,37], while the polyclonal test has a lower specificity and a tendency to overestimate the overall concentration of elastase [38].

Another indirect test to measure pancreatic exocrine function is the ^13^C-mixed triglyceride breath test, consisting of an oral administration of a ^13^C-marked test meal. The substrates are hydrolysed in proportion to the amount of pancreatic lipase activity. The breath samples reflect the absorption and metabolisation of products [39]. In contrast to the FE-1 test, this test can be used to assess the response to PERT treatment, as the breath results will normalise in the case of weight gain, normalisation of faecal fat, and nutritional deficiencies [40]. However, this test is only available in a limited number of centres in Europe and Asia and is not approved in the United States.

When dealing with the diagnosis of maldigestion, secondary to PEI in PC patients, it is important to keep in mind that different disease sites (head vs. body/tail of the pancreas), disease stage (local vs. advanced), and the received treatments (surgical resection of the pancreatic head or of the body/tail vs. none) result in different likelihoods of developing PEI. These factors can also affect the accuracy of the diagnostic test. Indeed, the post-test probability depends on the pre-test probability (which equals the prevalence of PEI) and on the accuracy of the diagnostic test. This means that, in conditions with a high prevalence of PEI, such as in patients who received resection of the pancreatic head, most diagnostic tests would not substantially increase the probability of diagnosing PEI. This is illustrated by the Fagan nomograms in Figure 2. Panel A shows the result from resected PC patients [27] and panel B is from locally advanced PC patients (LAPC) [41]. In both cases, the tumours were located in the pancreatic head. The sensitivity of FE-1 dosing is very high, but the specificity is rather low. In this scenario, the post-test probability of detecting PEI if FE-1 values are low (<200 µg/g) increases only moderately, from 67% to 74% in operated patients and from 87% to 90% in LAPC patients, respectively. Therefore, when the pre-test probability of PEI is very high and the accuracy of FE-1 is limited, the presence of symptoms or clinical and laboratory evidence of maldigestion should drive the decision to start PERT, irrespective of the result of a diagnostic test. However, in patients with resection of the pancreatic body or tail, dosage of FE-1 to investigate PEI before starting treatment might be more relevant. Indeed, Speicher et al. [42] evaluated FE-1 levels in PC patients operated with distal resection and reported that the probability of developing PEI was very low. The negative likelihood is higher, suggesting that if FE-1 levels are normal, the probability of detecting PEI becomes significantly lower (from 67% to 35% in operated patients and from 90% to 0% in non-operated patients). 

The incidence of PEI can also increase over the course of the disease. This was shown in a study by Sikkens et al., who studied 32 patients with advanced non-operable PC. The incidence of PEI was 66% at diagnosis, but increased to 92% at the end of the follow-up [43].

In summary, while many different diagnostic tests for PEI exist, a pragmatic approach might include: (a) Starting PERT in all patients with a pancreatic head resection or a tumour in the head of the pancreas and symptoms or signs of maldigestion; (b) testing for PEI by means of FE-1 levels in patients with an intermediate probability of PEI such as those with a tumour in the body or tail or those who underwent a distal pancreatic resection. The clinical suspicion of PEI, based on the typical symptoms and on signs of malabsorption and malnutrition, justifies the empirical use of PERT without prior testing [8].

## 4. Maldigestion and Treatment in Unresectable and Metastatic Pancreatic Cancer Patients

In metastatic PC (mPC), several mechanisms can contribute to PEI, such as the loss of pancreatic parenchyma and/or the obstruction of the main duct, which impedes the production of pancreatic enzymes or the transportation into the duodenum [40]. The concomitant impairment of bicarbonate secretion, necessary to buffer the gastric acid, can also lead to nutrient malabsorption [44]. Liver metastases can result in a liver impairment with reduced chemical digestion of fats and lipophilic vitamins [45]. On top of that, even if the patient did not undergo surgery, pancreatic fibrosis due to the desmoplastic reaction with the loss of angiogenesis can occur [46]. 

Patients with unresectable and metastatic PC had very bad prognoses, with a five-year overall survival of only 3% [47]. Therefore, the goals of PERT in this patient group is to increase the QoL, and if possible, survival. However, few studies investigated the impact of PERT on survival and/or QoL. In most studies, QoL was evaluated using EORTC QLQ-C30 with or without the PAN-26 module [48,49]. Table 1 gives an overview of the clinical trials on the use of PERT in this patient population. 

Certainly with the advent of new CT treatments within multidisciplinary teams, PERT is gaining importance. Indeed, only patients with good strength can cope with polychemotherapy treatments such as Folfirinox and the gemcitabine-nab-paclitaxel combination. This was shown in a retrospective study of 160 patients with unresectable PC [53]. The authors reported a significantly prolonged survival in patients treated with CT and PERT versus CT alone, and this effect was most pronounced in patients with significant weight loss at diagnosis. Furthermore, 13 patients who were initially considered unsuitable for CT could start a CT course after receiving supportive therapy, including PERT.

Several other studies have shown the benefits of PERT in patients with advanced PC, mostly in terms of weight gain [50,51,53] or even OS [53,55]. In addition, QoL is improved in patients receiving PERT [54], together with symptoms such as pancreatic and hepatic pain and diarrhoea. One study did not observe a statistical difference between PERT and placebo [51] regarding weight loss, QoL, or OS, but this could be attributed to the low dosage of PERT that was administered.

In conclusion, most of the unresectable and metastatic PC patients presented with symptoms of malabsorption and malnutrition or will have developed these over time. This could reduce QoL and/or performance status, impairing the possibility of receiving the aggressive CT treatments that could prolong survival. Despite most evidence being derived from retrospective studies or low sample trials, the use of PERT met their primary end point in advanced pancreatic cancer. In our opinion, PERT should be prescribed in almost all patients with mPC, as soon as the diagnosis is made.

## 5. Maldigestion in Locally Advanced/Borderline Setting

There is little data in the literature on the prevalence of PEI in patients with locally advanced (LA)/borderline resectable (BR) pancreatic cancer. In a small study, severe malabsorption of energy and fat was present in 50% of LA patients and it increased up to 80% after loco-regional ablative procedures. Poor intestinal absorption capacity and high faecal energy loss were due to increased faecal fat losses and were indirectly linked to PEI [56,57]. A systematic review of the literature identified two studies reporting PEI data in LA patients [58]. In the first study, patients were scheduled for curative resection, but were found to be LA at laparotomy. The authors reported a prevalence of PEI of 25% before surgery, which increased to 37% post-exploratory laparotomy [59]. The second study included patients identified as LA, either preoperatively or after laparotomy, and reported a preoperative prevalence of PEI of 50%; strikingly, overt steatorrhea was only reported in 10% of patients [60]. Despite the limited amount of data on PEI prevalence, PEI played an important role in this patient group: Neoadjuvant chemotherapy (NACT) may impair the functional reserve and lead to nutritional status changes. Additionally, weight loss and loss of muscle mass are limiting factors for CT choice, delivery, and tolerance, and may contribute to reducing a patient’s ability to undergo surgery. 

Several studies have attempted to document the impact of PERT on nutritional status in unresectable pancreatic cancer patients (including LA patients undergoing NACT), but the results are inconsistent. For instance, a small randomised study of 21 patients with unresectable cancer of the pancreatic head, who underwent biliary stent placement, showed a significant improvement in body weight, daily total caloric intake, and daily protein intake in the PERT group [61]. However, observational and randomised studies, including a variable proportion of LA patients (25–43%) failed to show a significant impact of PERT on nutritional parameters or survival outcomes [61,62]. Two other randomised studies, including small numbers of patients with LA disease, also failed to show differences in nutritional parameters, QoL, and/or survival for patients receiving PERT [51,52]. 

The results of studies evaluating the impact of PERT on oncological outcomes are also inconsistent. Pancreatic enzyme supplementation, as part of an alternative approach to PC therapy in stage II to IV patients, demonstrated no direct effect on survival when compared with gemcitabine monotherapy. In fact, patients in the latter group achieved longer survival and better QoL [63]. However, in a recent population-based study conducted in the UK [55], PERT was associated with a statistically and clinically significant survival advantage in PC patients when compared with matched, non-PERT-treated controls, and this was even despite the reported low usage of PERT (21.7%). Furthermore, survival was significantly greater among subjects receiving PERT, regardless of the studied subgroup with respect to the use of surgery or CT. In a retrospective analysis, CT and PERT usage were independently associated with longer survival in a model that also included the age and stage, again suggesting that the benefit of PERT may also apply to LA/BR patients undergoing NACT [55].

In a study specifically evaluating patients with BR PC undergoing NACT showed that only 43% of patients presenting with a higher baseline weight loss received PERT. The potential benefit of PERT on body composition, namely loss of fat-free mass and skeletal muscle mass during NACT (which were, in turn, associated with increased mortality), was not reported [64]. 

Overall, PEI was common in LA/BR patients who are candidates for NACT. The interplay between PEI and NACT in the complex: PEI results in malabsorption and malnutrition, while NACT, in its turn, may impair the functional reserve and induce nutritional status changes. However, weight loss and loss of muscle mass are limiting factors for CT and may influence fitness for surgery. Thus, PERT is a simple and attractive strategy to improve nutritional parameters and could have an impact on oncological outcomes, particularly in the NACT setting. However, solid evidence supporting its use is currently lacking and well-designed; adequately sized trials are urgently needed in this setting.

## 6. Maldigestion in Resected Pancreatic Cancer 

A considerable number of resected cancer patients develop PEI. The severity of post-surgical PEI depends on the underlying disease, the preoperative pancreatic function, and the extent of the resection [65]. The management of PC relies on a combination of surgery and CT. The latter can be given before (neoadjuvant) and/or after (adjuvant) surgery. According to the location and extent of the disease, pancreaticoduodenectomy (PD) and distal pancreatectomy (DP) are the most frequently performed surgical procedures. Patients undergoing PD for PC developed PEI in 64–100% of cases, whereas this percentage decreased to 0–42% after DP [66,67,68,69]. Total pancreatectomy (TP) is a less frequent procedure, mainly performed in the presence of advanced PC (especially when requiring arterial resection) [70] or in the presence of PC associated with a intraductal papillary mucinous neoplasm. As expected, 100% of patients submitted to TP developed a severe endocrine and exocrine pancreatic insufficiency. 

The presence of PEI should be evaluated at the time of diagnosis, particularly in those patients with borderline resectable pancreatic cancer who are candidates for NACT. PERT is helpful to improve their nutritional and general health status, thus enabling these patients to undergo NACT and subsequently major surgery [55,71]. The pancreatic exocrine function should also be evaluated after surgery and eventually supplemented with PERT to ensure proper postoperative recovery, allowing the patient to undergo adjuvant CT. 

Objective diagnosis of PEI after pancreatic resection is difficult as the available pancreatic function tests are not accurate or need further validation in this particular setting. FE-1, which is the most common test used in clinical practice, appeared to be unreliable in these patients [67]. The validity of another test, the ^13^C-labeled mixed triglyceride breath test, is still a matter of debate, as it needs further validation [72]. Hence, a straightforward approach for PEI diagnosis after pancreatic surgery is lacking. Therefore, PERT treatment should not only start when a test diagnoses PEI, but also when there is clinical suspicion of PEI (steatorrhea, weight loss and maldigestion-related symptoms). In this case, the improvement of symptoms and nutritional markers after an empiric therapeutic trial with PERT can be a substitute for an official diagnosis of PEI [71]. Asymptomatic patients with an abnormally high daily faecal fat excretion are also candidates for PERT, as they are at high risk of developing nutritional deficits [73]. Finally, all the patients who are candidates for PD should be considered at high risk for PEI, regardless of the underlying disease. Patients with PC should be regarded as patients with an even higher risk, as this malignancy is frequently associated with obstructive chronic pancreatitis and fibrosis/atrophy of the parenchyma, which are risk factors for PEI development by themselves. Therefore, it has been suggested to give PERT to all patients submitted to PD for PC, especially those who will undergo adjuvant CT [27,65]. 

The difficulty to diagnose PEI in resected patients can have severe consequences: PEI can remain unnoticed or is under-diagnosed, with patients not receiving the right dosage of PERT [11]. In the end, this will result in weight loss, frailty, and the inability to recover from major surgery or to tolerate CT [55]. Dose titration and monitoring should be considered on an individual basis in the long term. Dose optimisation of PERT is necessary for effective management of PEI, in addition to the regular evaluation of nutritional status, appropriate patient education, and reassessment whenever symptoms return [74].

PERT with pancrelipase in the form of delayed-release capsules containing enteric-coated minimicrospheres is the cornerstone in the treatment of PEI following pancreatic surgery [75,76]. PERT has been reported to be associated with a significant improvement of fat and protein digestion compared to the placebo in patients submitted to pancreatic resection. Moreover, PERT also improved the clinical manifestations of PEI and QoL [66,77].

Recently, a possible role of PEI in survival has been reported in the setting of major pancreatic surgery followed by adjuvant CT [55]. This is not surprising, as untreated PEI, resulting in malnutrition and frailty, may lead to decreased survival [71]. Finally, PERT is generally well-tolerated at the high doses administered, it has a relatively low cost, and it can be delivered by non-specialists in the community.

## 7. Pancreatic Enzyme Replacement Therapy: Tips and Tricks

### 7.1. Dosage Challenges

The available formulations contain pancreatic enzymes encapsulated in microgranules or minimicrospheres with a pH sensitive coating. This coating prevents the release and the subsequent inactivation of the enzymes by gastric acid and promotes the release of the enzymes into the intestinal lumen where the pH is higher and optimal for the digestion and absorption of food [65].

The initial recommended dose of pancreatic extract is 40,000–50,000 U.Ph.Eur of lipase per meal and 25,000 U.Ph.Eur per snack. The dose should be increased until the steatorrhea is sufficiently reduced [78,79]. When this is achieved, the dosage should be maintained. The patients should be educated to modify the quantity of pancreatic extracts themselves to resolve the steatorrhea, as a third–half of patients will require individualised treatment [80]. In the case of an unsatisfactory response, the amount of lipase can be increased by ±20,000 U.Ph.Eur during meals, even though the data do not show that an increased dose is associated with an increased effectiveness of pancreatic extracts [80].

### 7.2. Dietary and Drug Recommendation

Food intake should be distributed, if possible, between three main meals per day (breakfast, lunch, and dinner), and two or three snacks. The pancreatic extracts should be ingested during the meals, rather than before or after the meal. The caloric intake should not be restricted [65].

A diet rich in fibre is contraindicated, as the fibrous material can interfere with proteolytic and amylolytic enzyme activity; lipolytic activity is affected the most [81]. Enzymes contained in gastroprotected minimicrospheres can be consumed with food, having a pH < 5.5. Acid-suppressing agents should only be taken by patients who continue to experience symptoms of maldigestion despite the adequate administration of PERT [82] or in the presence of other upper gastrointestinal symptoms [83]. 

### 7.3. Goal of the Treatment

Steatorrhea in severe pancreatic insufficiency is very difficult to resolve completely. Only a 60–70% reduction is usually achieved using PERT [84] because of the numerous interactions between pancreatic maldigestion, intestinal ecology, and intestinal inflammation. Fat-soluble vitamins and micronutrients, such as zinc and selenium, should be routinely assessed and administered whenever necessary [85].

### 7.4. Warnings Regarding PERT

Crushing, chewing or holding the pancreatic extract capsules in the mouth may cause local irritation. The extracts should only be administered without gelatinous capsule to patients with gastrectomy [65].

## 8. Conclusions

PEI is a hallmark of PC, although its prevalence and severity depends on a number of factors, such as the location of the tumour (tumours in the head of the pancreas are associated with a higher prevalence of PEI than tumours in the body/tail) and the type of surgery the patient undergoes in the case of resectable PC (patients undergoing PD for PC develop PEI in 64–100% of cases, whereas this percentage decreases to 0–42% after DP) [68]. However, in common clinical practice, PEI is frequently not detected and appropriate PERT is underused (21.7% in a UK population-based study [55]).

With this position paper, we aimed to increase the awareness on the importance of PEI and malnutrition in all groups of PC patients, ranging from unresectable and metastatic patients, to locally advanced and borderline patients, and to resectable patients. As a consequence of PEI, these patients often face severe weight loss and cachexia, resulting in frailty that excludes them from undergoing surgery or CT, a decreased QoL and even a shorter OS. Therefore, we recommend that every physician treating PC patients, being an oncologist, surgeon, or gastroenterologist, should look for signs of malnutrition, and in the case of suspicion, refer the patient to a clinical nutrition specialist to implement prophylactic nutritional monitoring and/or counselling. During counselling, patients should be guided into self-management of PERT, in order to learn how to optimise PERT to their needs and their diet. 

In addition to creating awareness, we wanted to provide practical information on malnutrition, PEI and PERT. A pragmatic approach to test and treat patients with pancreatic cancer is provided in Figure 3. In Table 2, we summarise the major key points of each chapter.

## Figures and Tables

**Figure 1 cancers-12-00275-f001:**
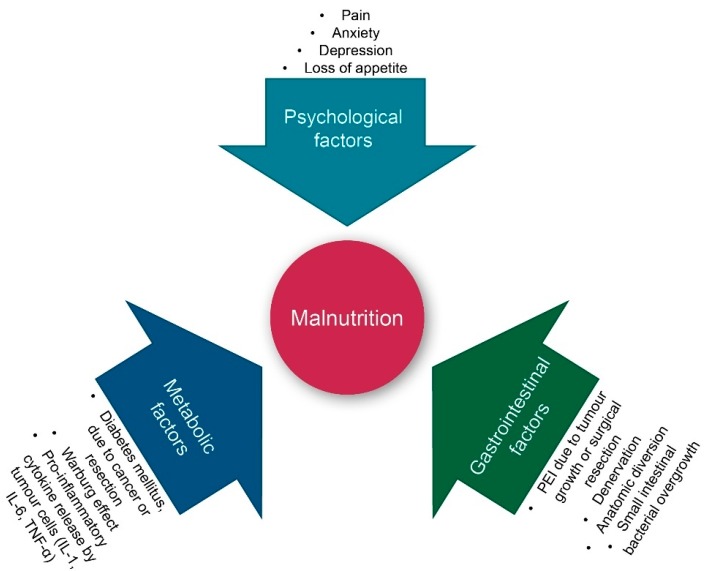
Factors contributing to malnutrition in pancreatic cancer patients. IL: interleukin; PEI: pancreatic exocrine insufficiency; and TNF-α: tumour necrosis factor alpha. Warburg effect: See the text for an explanation.

**Figure 2 cancers-12-00275-f002:**
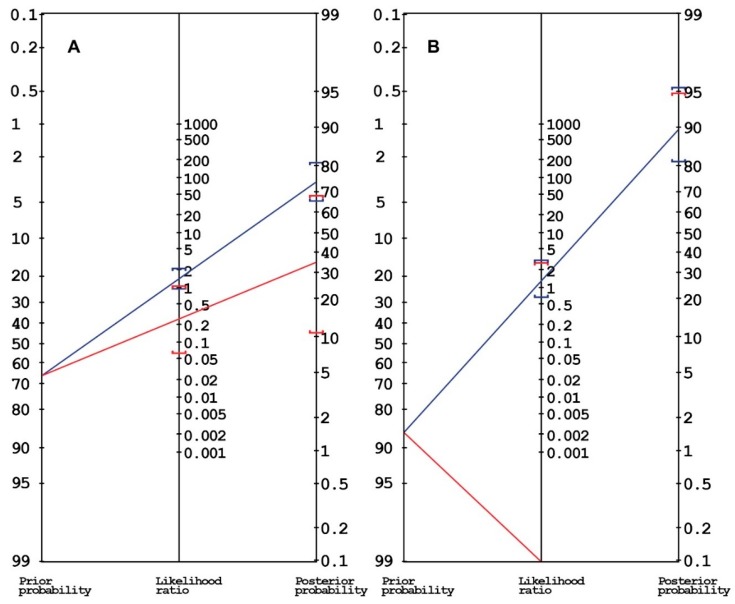
Panels A and B represent examples of Fagan nomograms showing how the probability of diagnosing pancreatic exocrine insufficiency (PEI) by means of faecal elastase 1 (FE-1) dosage can vary in different scenarios, depending on its prevalence and on the accuracy of the test. On the left: Data from 40 operated patients, with a pre-test probability of having PEI of 67%. The FE-1 test has a sensitivity of 91%, a specificity of 35% and a positive likelihood ratio (+LR) of 1.4. The post-test probability increases only moderately to 74% (blue line) [27]. The negative likelihood ratio (−LR) of 0.26 suggests that, in the case of normal FE-1 levels, the post-test probability of PEI would be as low as 35% (red line). On the right (panel B): The case of 15 unoperated, locally advanced pancreatic cancer patients, with a pre-test probability of 87%. In this case, the post-test probability would only increase to 90% when FE-1 levels are reduced (blue line), but would decrease to 0% (red line) in the case of normal values (sensitivity 100%, specificity 22%, +LR 1.28, −LR 0) [41].

**Figure 3 cancers-12-00275-f003:**
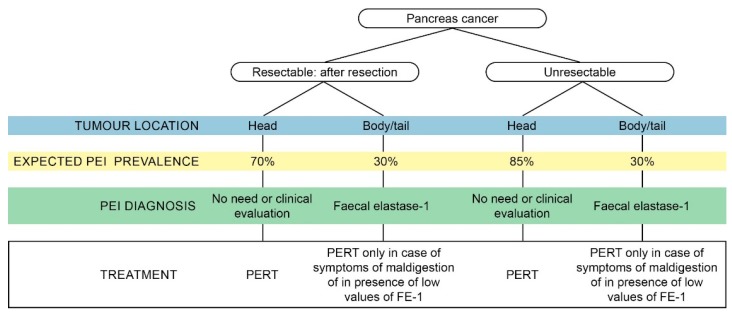
Pragmatic approach to testing and treating pancreatic exocrine insufficiency (PEI) in patients with pancreatic cancer.

**Table 1 cancers-12-00275-t001:** List of studies investigating the effect of PERT in metastatic pancreatic patients.

References	Study Type	Treatment	Outcome
[50]	Randomised	Panzytrat * 50,000 U.Ph.Eur/meal–25,000 U.Ph.Eur/snack	Improved nutritional status in Panzytrat-group: increase body weight (1.2% vs. body weight loss of 3.7%), increased CFA (12%). The daily total energy intake was 8.42 MJ and 6.66 MJ in pancreatic enzymes-treated group and in placebo patients, respectively (*p* = 0.04, 95% CI 0.08–3.44).
		Placebo	
[51]	Randomised	Norzyme^®^ ** 2 capsules/meal–1 capsule/snack	No significant difference in body weight change, PG-SGA score, QoL or OS.
		Placebo	
[52]	Randomised	Creon^® £^	Greater weight loss in placebo-group, no statistical difference in BMI, nutrition score, QoL, mOS (mOS was 67.6 (95% CI 14.1–98.4) weeks and 17 (95% CI 8.1–48.7))
		Placebo	
[53]	Retrospective	Creon^® £^ 50,000 U.Ph.Eur/meal–25,000 U.Ph.Eur/snack	Significant longer survival in Creon-group than placebo (189 days (95% CI 167.0–211.0 days) vs. 95.0 days (95% CI 75.4–114.6 days respectively (HR: 2.117, 95% CI 1.493–3.002; *p* < 0.001))
		Placebo	
[54]	Prospective	50,000 IU Creon/meal–25,000 IU Creon/snack 1 extra 25,000 IU Creon per 16–20 g of extra fat/meal or snack	Significant reduction of pancreatic and hepatic pain compared before the treatment (47 vs. 33 and 24 vs. 11, respectively, *p* < 0.05) and diarrhoea scores (26 vs. 8, *p* < 0.005). Significant improvement of QoL.
[55]	Retrospective population-based	PERT–dosage not specified	Longer survival in PERT-group respect placebo group in both patients chemotherapy treated (328 days and 226 days, respectively, *p* < 0.0006) or in best supportive care (171 days and 71 days, respectively, *p* < 0.001).

* Panzytrat: 25,000 U.Ph.Eur lipase + 1250 U.Ph.Eur proteases + 22,500 U.Ph.Eur amylase/capsule. ** Norzyme: 25,000 U.Ph.Eur lipase + 1250 U.Ph.Eur proteases + 22,500 U.Ph.Eur amylase/capsule. ^£^ Creon: 25,000 BP U lipase + 18,000 BP U amylase + 1000 U.Ph.Eur protease/capsule. BMI: body mass index; CFA: coefficient of fat absorption; CI: confidence interval; CT: chemotherapy; HR: hazard ratio; mOS: median overall survival; MJ: Megajoule; PERT: pancreatic enzyme replacement therapy; PG-SGA: patient-generated subjective global assessment; and QoL: quality of life.

**Table 2 cancers-12-00275-t002:** Take home messages regarding PEI and PERT in pancreatic cancer patients.

**Nutritional Support Principles in Pancreatic Cancer**
Early nutritional support, starting with individualised dietary counselling and possibly implementing artificial nutrition, is mandatory in all pancreatic cancer (PC) patients at nutritional risk, as it is able to improve clinical outcomes and reduce treatment complications.
Considering the high prevalence of nutritional derangements in PC, we recommend to refer every patient with PC to a clinical nutrition specialists for implementing prophylactic nutritional monitoring and/or counselling.
Dietary advice should be tailored to the individual patient and “hypocaloric alternative anticancer diets” and “natural nutrients” not supported by clinical evidence should be avoided.
**Real life diagnostic approach to maldigestion**
Although a validated “gold standard” method to assess pancreatic enzyme insufficiency (PEI) is lacking, faecal elastase is generally used in clinical practice, as it is promptly available and least invasive.
Treatment of PEI using pancreatic enzyme replacement therapy (PERT) should start as soon as PEI is diagnosed (even if the patient is asymptomatic) or when a high clinical suspicion of PEI is present.
In patients with the tumour located at the head of the pancreas, the prevalence of PEI is so high that all patients should be treated with PERT, even without testing.
**Maldigestion in unresectable and metastatic pancreatic cancer**
Despite that large randomised clinical trials are missing, the data in literature indicate that PERT can enhance nutritional status, allowing the patient to undergo chemotherapy (CT), increase quality of life (QoL) and overall survival (OS).
**Maldigestion in borderline/locally advanced setting**
There is a big gap on information regarding the prevalence of PEI in the borderline/locally advanced setting, and the potential of PERT in this setting.
Malnutrition is important in the neoadjuvant setting: on the one hand, neoadjuvant chemotherapy (NACT) may impair the functional reserve and lead to nutritional status changes. On the other hand, weight loss and loss of muscle mass are limiting factors for CT choice, delivery, and tolerance and contribute to reducing a person’s ability to undergo surgery.
Locally advanced (LA)/borderline resectable (BR) pancreatic cancer patients should always be assessed for PEI and nutritional status before starting NACT, closely monitored during treatment, and supported with PERT and nutritional counselling as appropriate.
**Maldigestion in resected pancreatic cancer**
Patients submitted to pancreaticoduodenectomy (PD) should be considered at high risk for developing PEI, especially in the presence of PC. We recommend to start PERT routinely in these patients, especially in those who will undergo adjuvant CT.
Most PC patients suffering from PEI are undertreated which can result in malnutrition and frailty, reducing patient’s ability to undergo major surgery and CT. An adequate treatment of PEI is therefore essential for patients affected by resectable PC, both before surgery, especially if they undergo NACT, and after surgery, to guarantee a proper postoperative recovery and the capacity to tolerate adjuvant CT.
**Pancreatic enzyme replacement therapy: tips and tricks**
The initial recommended dose of pancreatic extract which should be given is 40,000–50,000 U.Ph.Eur of lipase per meal and 25,000 U.Ph.Eur per snack, and this dose should be increased until the steatorrhea is sufficiently reduced. This dosage should be maintained over time. Dose optimisation of PERT is necessary for an effective management of PEI, in addition to regular evaluation of nutritional status, appropriate patient education and reassessment whenever symptoms return.
Food intake should be distributed between three main meals per day and two or three snacks. The pancreatic extracts should be ingested during the meals. The caloric intake should not be restricted.
A diet rich in fibre is contraindicated because the fibrous material will interfere with enzyme activity
Crushing, chewing or holding the pancreatic extract capsules in the mouth may cause local irritation.

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
