# Peer review of "Pancreatic Enzyme Replacement Therapy in Pancreatic Cancer"

_cancers, 2020, doi:10.3390/cancers12020275_

Round 1
Reviewer 1 Report
Pezzilli et al have written a comprehensive excellent review on pancreatic enzyme replacement therapy (PERT) and it's benefits in the prognosis of pancreatic cancer, in patients undergoing chemotherapy. The authors have addressed the importance of treating malnutrition and enzyme insufficiency pancreatic cancer. Then they provided PERT option for each subset of patients based on their clinical experience.
I have only minor comments
The sentence "Two oncologists, two gastroenterologists, 25 one nutritionist and one pancreatic surgeon have reviewed" in the abstract sounds unprofessional. The specializations of authors should be clear from the author affiliations. Is there an overall probability number of percent of pancreatic cancer patients that present with PEI at diagnosis before chemo? How is QoL measured? Do you think changing the parameters can influence the outcome of some the trials reported for mPC and locally advanced patients on PERT? Several grammatical and sentence construction errors in English. The writing needs improvement.Author Response
The sentence "Two oncologists, two gastroenterologists, 25 one nutritionist and one pancreatic surgeon have reviewed" in the abstract sounds unprofessional. The specializations of authors should be clear from the author affiliations.We agree with the comment of the reviewer. The sentence has been modified accordingly (page 1, Line 6). Is there an overall probability number of percent of pancreatic cancer patients that present with PEI at diagnosis before chemo?
Very little is known about the prevalence of PEI in patients before chemo. On page 7, Lines 245-248, we report on a small scale study where severe malabsorption was present in about half of locally advanced pancreatic cancer patients and increased up to 80% after loco-regional ablative procedures. How is QoL measured?
We thank the reviewer for this comment. QoL has been measured by EORTC QLQ-C30 module with or without PAN-26 module. This information has been added to the text and referenced (page 6, lines 215-216, references 48-49). Do you think changing the parameters can influence the outcome of some the trials reported for mPC and locally advanced patients on PERT?
We thank the reviewer for this challenging comments, but at the moment data is lacking to provide an answer to this question. Several grammatical and sentence construction errors in English. The writing needs improvement.
A native speaker has extensively revised the English language.
Reviewer 2 Report
Thank you for the opportunity to review your work. I have a few major and minor comments regarding your work.
Major comments:
All sections have too many repetitions and tend to deviate from the principal of the paper. Please revise the style of writing and recalibrate the paper towards the common goal of eliciting guidelines. Line 117-121- Authors support that parenteral nutrition is better than enteral nutrition. A little bit more discussion is needed into the advantages versus disadvantages of the two modalities. The section on pancreatic enzyme replacement therapy is very short. In fact, for an article that promises to talk about the pancreatic enzyme replacement therapy, the reader does not really get much information about this challenging aspect on how to counter PEI. I would suggest the authors read the following article 25206255 and incorporate changes in their manuscript. This article provides some good insights into the use of pancreatic enzyme replacement therapy in the 21st century. The four sections divided on how PEI affects the 4 different kinds of pancreatic cancer can be clubbed into two main sections- resectable and unresectable/metastatic pancreatic cancer. A locally unresectable pancreatic cancer, rarely converts into a resectable one, even after neoadjuvant chemotherapy. As authors acknowledge, more than 80% of patients present with metastatic disease, hence, we can deduce that other stages are not encountered as commonly. I would also urge the authors to discuss the direct tests for PEI. This being a review article should be comprehensive in including everything that is currently, either in the developmental stage or in clinical practice that helps in determining PEI.Minor comments:
The abbreviation PEI is denoting two different words- different expansion in abstract and in the introduction. Please choose one. The abbreviation PERT has not been expanded in the abstract or keywords. Please do so. Line 25- The line in the abstract is confusing and may convey that it is easy to identify PEI when I believe authors want to convey that it is not easy to do so. Line 34-39 can be compressed in 1-2 sentences. The article is quite long and should provide a pointed overview. Line 50-51 is a repetition. Please omit. Please use words that are more apt for academic writing. For example: in figure 1- Instead of using the word sorrow, depression is more apt in this setting. I would suggest using another abbreviation for the pancreatic function test. PFT is quite popular for pulmonary function tests and can create confusion.Author Response
Major comments:
All sections have too many repetitions and tend to deviate from the principal of the paper. Please revise the style of writing and recalibrate the paper towards the common goal of eliciting guidelines.The reviewer is under the impression that the goal of the paper was to elicit guidelines. That is not the case, given that there is not a lot of evidence of high quality on PERT in PEI. Instead, the paper aims to provide practical recommendations, based on the data in literature and on the authors’ own experience (as we have explained in the abstract and the introduction of the paper). The practical recommendations are put together in a table, so that they can easily be found if someone does not have the time to go through the paper. Line 117-121- Authors support that parenteral nutrition is better than enteral nutrition. A little bit more discussion is needed into the advantages versus disadvantages of the two modalities.
We have slightly rewritten this section to provide more clarity (page 3, Lines 112-119). However, an extensive discussion on the advantages of disadvantages of parenteral/enteral nutrition is outside the scope of the article. The provided references can be consulted in case more information is desired. The section on pancreatic enzyme replacement therapy is very short. In fact, for an article that promises to talk about the pancreatic enzyme replacement therapy, the reader does not really get much information about this challenging aspect on how to counter PEI. I would suggest the authors read the following article 25206255 and incorporate changes in their manuscript.This article provides some good insights into the use of pancreatic enzyme replacement therapy in the 21st century.
We thank the reviewer for the suggestion. We have modified the text (page 9, Lines 348-355). The four sections divided on how PEI affects the 4 different kinds of pancreatic cancer can be clubbed into two main sections- resectable and unresectable/metastatic pancreatic cancer. A locally unresectable pancreatic cancer, rarely converts into a resectable one, even after neoadjuvant chemotherapy. As authors acknowledge, more than 80% of patients present with metastatic disease, hence, we can deduce that other stages are not encountered as commonly.
We apologize, but we do not agree with this comment. Even if the major part of pancreatic cancer patients are diagnosed in advanced stage of the disease, the actual chemotherapy approach is to treat all available patients in order to downstage the cancer. Thus, we believe the various section are appropriate. I would also urge the authors to discuss the direct tests for PEI. This being a review article should be comprehensive in including everything that is currently, either in the developmental stage or in clinical practice that helps in determining PEI.
We thank the reviewer for this comment that improve the quality of our manuscript. We have added a section on direct tests on page 4, Lines 136-144.
Minor comments:
The abbreviation PEI is denoting two different words- different expansion in abstract and in the introduction. Please choose one.We apologize for the error. We have updated the text so that PEI is consistently referred to as “pancreatic exocrine insuffiency”. The abbreviation PERT has not been expanded in the abstract or keywords. Please do so.
We apologize for the error. The abstract and keywords were updated to address the problem. Line 25- The line in the abstract is confusing and may convey that it is easy to identify PEI when I believe authors want to convey that it is not easy to do so.
We have updated the abstract to address this comment (abstract section, Lines 24-25). Line 34-39 can be compressed in 1-2 sentences.
We have updated the text to address this remark (page 1, Lines 35-39). The article is quite long and should provide a pointed overview. Line 50-51 is a repetition. Please omit.
We have updated the text to address this remark. Please use words that are more apt for academic writing. For example: in figure 1- Instead of using the word sorrow, depression is more apt in this setting.
We have updated figure 1 to address this remark. I would suggest using another abbreviation for the pancreatic function test. PFT is quite popular for pulmonary function tests and can create confusion.
Thank you for this comment. We have updated the text so that the abbreviation was no longer needed.
Reviewer 3 Report
A well written and much needed review of the literature in this field. This article flows well and addresses most of the key issues for this patient group.
Would benefit from a little more explanation of the Warburg effect (stated as "explained in the text" on line 69, but only a very brief mention on line 44.).
Author Response
A well written and much needed review of the literature in this field. This article flows well and addresses most of the key issues for this patient group.
We thank the reviewer for the comments on our manuscript.
Would benefit from a little more explanation of the Warburg effect (stated as "explained in the text" on line 69, but only a very brief mention on line 44.).
We thank the reviewer for the comments that improve the quality of our manuscript. The text on Warburg effect has been modified (Page 1, line 44 and page 2, lines 45-47).
Round 2
Reviewer 2 Report
Thank you for the opportunity to review the paper.
Substantial changes have been made to the manuscript which makes it more helpful and adaptable for a reader in real life.